# Exposure of Bladder Cancer Cells to Blue Light (λ = 453 nm) in the Presence of Riboflavin Synergistically Enhances the Cytotoxic Efficiency of Gemcitabine

**DOI:** 10.3390/ijms25094868

**Published:** 2024-04-29

**Authors:** Sofia Sturm, Günter Niegisch, Joachim Windolf, Christoph V. Suschek

**Affiliations:** 1Department of Orthopedics and Trauma Surgery, Medical Faculty, University Hospital Düsseldorf, Heinrich-Heine-University Düsseldorf, Moorenstr. 5, 40225 Düsseldorf, Germany; sofia.sturm@hhu.de (S.S.); windolf@hhu.de (J.W.); 2Department of Urology, Medical Faculty, University Hospital Düsseldorf, Heinrich-Heine-University Düsseldorf, 40225 Düsseldorf, Germany; guenter.niegisch@med.uni-duesseldorf.de

**Keywords:** bladder tumour, urothelial cancer, chemotherapy, mitomycin C, blue light, reactive oxygen species, cell death, gemcitabine, riboflavin, cytotoxicity, oxidative stress, BFTC-905, SW-1710, RT-112

## Abstract

Non-muscle invasive bladder cancer is a common tumour in men and women. In case of resistance to the standard therapeutic agents, gemcitabine can be used as off-label instillation therapy into the bladder. To reduce potential side effects, continuous efforts are made to optimise the therapeutic potential of drugs, thereby reducing the effective dose and consequently the pharmacological burden of the medication. We recently demonstrated that it is possible to significantly increase the therapeutic efficacy of mitomycin C against a bladder carcinoma cell line by exposure to non-toxic doses of blue light (453 nm). In the present study, we investigated whether the therapeutically supportive effect of blue light can be further enhanced by the additional use of the wavelength-specific photosensitiser riboflavin. We found that the gemcitabine-induced cytotoxicity of bladder cancer cell lines (BFTC-905, SW-1710, RT-112) was significantly enhanced by non-toxic doses of blue light in the presence of riboflavin. Enhanced cytotoxicity correlated with decreased levels of mitochondrial ATP synthesis and increased lipid peroxidation was most likely the result of increased oxidative stress. Due to these properties, blue light in combination with riboflavin could represent an effective therapy option with few side effects and increase the success of local treatment of bladder cancer, whereby the dose of the chemotherapeutic agent used and thus the chemical load could be significantly reduced with similar or improved therapeutic success.

## 1. Introduction

Bladder carcinoma (BC) is a frequent cancer of the urinary tract, accounting for more than 550,000 new cases every year [1,2]. While 25% of patients suffer from muscle-invasive bladder cancer (MIBC), having a high risk of metastatic disease with poor overall prognosis, often requiring extensive multimodal therapy, most patients present with a non-muscle-invasive bladder cancer (NMIBC) [3,4,5]. Whereas metastatic disease is only rarely observed in these patients, recurrences requiring endoscopic therapy (transurethral resection) are frequent, and a significant number of patients are prone to undergo tumour progression to MIBC [5,6]. Though instillation therapy using chemotherapeutics (e.g., mitomycin C, MMC) or the immunotherapeutic bacillus Calmette–Guérin (BCG) is recommended to decrease the risk of recurrence and progression, 5 to 25% will recur and potentially progress to MIBC despite adequate adjuvant therapy [5]. From a biological point of view, bladder cancer cells of NMIBC patients progressing to muscle-invasive disease have molecular alterations similar to those observed in MIBC (loss of retinoblastoma gene RB1, E2F3 transcription factor overexpression, cyclin-dependent kinase inhibitor 2A deletion, p53 mutation) [7].

Moreover, instillation therapy with BCG or MMC may exhibit significant side effects. Cystitis, fever [8], leukocyturia [9], tuberculosis-like illness and granulomatous prostatitis [10] can occur during treatment with BCG, whereas cystitis-like symptoms such as urgency, dysuria, increased micturition or local bladder wall reactions and allergic reactions of the skin can occur during treatment with MMC [8,10]. These therapies can therefore be of high risk for patients in a certain age group or with pre-existing conditions and could lead to lower compliance [11]. Furthermore, BCG [12,13] or MMC [14,15] are not effective in all patients, thus new options need to be explored. The aim should therefore be to develop alternative new forms of therapy in which better therapeutic success can be achieved through the use of various innovative measures while at the same time limiting the side effects, for example by reducing the therapeutic dose of the respective cytostatic agent.

Gemcitabine is used as salvage instillation therapy in BCG refractory patients who are not eligible for or refuse to undergo radical cystectomy [16,17,18,19,20]. Blue light has the property of reducing flavins, which can lead to a significantly increased production of reactive oxygen species, whereby flavin-containing enzymes or exogenously applied flavin derivatives can serve as the source for the ROS-generating flavins [21,22,23]. Blue light has already been evaluated in cancer treatment of fibrosarcoma [24], leukaemia [25], melanoma [26,27], skin tumour [28] and colon cancer [24,29] and is used in the treatment of acne [30], actinic keratoses [31], atopic dermatitis [32], plaque psoriasis [33] and icterus neonatorum [34]. Very recently, we could demonstrate that blue light exposure (453 nm) of human urothelial bladder carcinoma cells notably enhanced the therapeutic efficiency of mitomycin C in the form of significantly enhanced cytotoxicity via apoptosis and secondary necrosis [35]. Due to these properties, blue light might represent an effective, low-side-effect and success-enhancing therapy option in the local treatment of bladder cancer.

Combining blue light with flavin-containing molecules has been observed to increase the production of reactive oxygen species (ROS) [21,22], which enhances apoptosis and thus DNA damage, leading to cell death [23]. Riboflavin is a part of the vitamin B2 complex and an important component in living organisms, for example, as flavin adenine dinucleotide (FAD) in the respiratory chain [36]. The apoptotic effect of irradiated riboflavin could be shown for human leukaemia cells (HL60) [37,38], human prostate cancer cells (PC3) [39], mouse melanoma cells (B16F10) [40] and renal carcinoma cells (786-O) [41]. For the therapy of melanoma, it has even been shown that photodynamic therapy with blue light and riboflavin delivers good results in vivo [42].

In our preclinical in vitro study, we characterised the influence of exposure to blue light at 453 nm in the presence of the photosensitiser riboflavin on the therapeutic efficacy of gemcitabine in three molecularly defined urothelial carcinoma cell lines (BFTC-905, SW-1710, RT-112). Further, we investigated the underlying molecular mechanisms.

## 2. Results

### 2.1. Evaluation of the Cytotoxic Capacity of Gemcitabine

Urothelial carcinoma cells growing in the form of a nearly confluent monolayer were incubated with 10, 25 or 50 ng/mL gemcitabine, and, after 24 h of incubation, the number of living cells was detected by CTB assay.

The results of this series of experiments, as shown in Figure 1, confirm a gemcitabine concentration-dependent increase in cell toxicity. With 10 ng/mL gemcitabine, viability was significantly decreased in RT-112 only (−10%) (Figure 1B), while we could not observe any significant influence on the viability of BFTC-905 (Figure 1A) and SW-1710 (Figure 1C) cell cultures. With 25 ng/mL gemcitabine, we observed a clear and statistically significant toxicity in the cells of all three tumour cell lines. We could again observe the strongest loss of vitality in the RT-112 cultures (approx. 40–60%), followed by the BFTC-905 cultures (30–60%) and the SW-1710 cell line (10–30%). As expected, we were able to achieve the highest toxicity rates in all three cell lines with 50 ng/mL gemcitabine. Toxicity values of 50–80% were quantified here, whereby the toxicity values in the different tumour cell cultures did not differ significantly.

### 2.2. Impact of Blue Light Exposure on Viability of Urothelial Carcinoma Cells

Next, the dose–response of blue-light irradiation was evaluated in the UC cell lines. As part of the evaluation of a suitable light dose of blue light, urothelial carcinoma cells, which grew in the form of an almost confluent monolayer, were irradiated with increasing light doses, and, after 24 h of incubation in the incubator, cell viability was quantified using MTT assay.

Exposure of the urothelial tumour cell lines led to a light dose-dependent increase in cytotoxicity, the course of which was almost identical in the three cell lines examined (Figure 2). A reproducible and statistically significant increase in toxic events in all three tumour cell lines used was observed after exposure to a light dose of 140 J/cm^2^. In accordance with our specifications, as a result of this test series, we decided to use blue light in a light dose of 110 J/cm^2^ in all subsequent test series. We decided to use this subtoxic dose to distinguish better between the effect of blue light and the other substances we were using during the following experiments.

### 2.3. Presence of Riboflavin Enhances Blue Light-Induced Cytotoxicity on Urothelial Carcinoma Cells

Urothelial carcinoma cells growing in the form of a nearly confluent monolayer on 12-well cell culture plates were exposed to blue light (110 J/cm^2^). In addition, cells were incubated with 0 (control), 5, 10, 20 or 25 µM riboflavin. After 24 h of incubation, cell viability was evaluated by MTT assay.

The exposure of urothelial carcinoma cell lines to blue light (110 J/cm^2^) did not result in any statistically significant changes in cell vitality in the exposed cell lines (Figure 3). However, cytotoxic events were clearly demonstrated in cell cultures exposed to blue light in the presence of riboflavin. In SW-1710 and BFTC-905 cultures, exposure to blue light in the presence of 10, 20 or 25 µM riboflavin resulted in a statistically significant increase in cytotoxicity compared with the values observed after irradiation of the respective cell cultures in the absence of riboflavin (Figure 3). Only in RT-112 cultures with 20 or 25 µM riboflavin, a statistically significant reduction in cell viability was found. Since, in these series of experiments, the lowest effects on cell vitality with light-exposed cultures was observed with 10 µM riboflavin, we used this riboflavin concentration in the following experiments to determine possible additive or synergistic effects.

### 2.4. Impact of Blue Light-Activated Riboflavin on Cytotoxic Capacity of Gemcitabine

Urothelial carcinoma cells growing as nearly confluent monolayer on 12-well cell culture plates were incubated with 10 or 25 µM gemcitabine. Additionally, gemcitabine- treated cultures were exposed to non-toxic doses of blue light (110 J/cm^2^) in the presence of 10 µM riboflavin. After 24 h of incubation, the number of living cells in the control and treated cultures was detected by MTT assay, as described above.

The data presented in Figure 4 allow the conclusion that additional exposure of gemcitabine-treated tumour cultures to blue light significantly increased the cytotoxic capacity of the drug. Here, treatment with 10 µM gemcitabine plus exposure to blue light led to cytotoxicity levels that were comparable to those observed using 25 µM gemcitabine alone. The latter treatment with 10 µM gemcitabine plus 110 J/cm^2^ of 453 nm but carried out in the presence of 10 µM riboflavin led to further potentiation of the cytotoxicity of gemcitabine towards the urothelial tumour cell lines. The treatment mode with the triple combination resulted in a cytotoxicity rate that was statistically significantly higher than that of 10 µM gemcitabine and gemcitabine plus blue light. With almost 75% culture toxicity, the cytotoxic potential of this type of treatment was even significantly higher than we observed with treatment with 50 µM gemcitabine alone (Figure 1).

### 2.5. Impact of Blue Light Exposure in the Presence and Absence of Riboflavin on Lipid Peroxidation and Mitochondrial Respiratory Chain Activity of Urothelial Carcinoma Cells

In search of a molecular mechanism of the above observations, we evaluated the influence of exposure of urothelial tumour cell lines to blue light in the absence and presence of riboflavin on the degree of achieved lipid peroxidation and activity modulation of the mitochondrial respiratory chain.

The results presented in Figure 5A clearly indicate that, regardless of the gemcitabine concentration present in the cell culture, exposure of cell cultures of the urothelial tumour cell line BFTC-905 to blue light alone (110 J/cm^2^) resulted in a statistically significant 2–3-fold increase in lipid peroxidation rate compared with the unirradiated cell cultures. Blue light irradiation of BFTC-905 cultures in the presence of 10 µM riboflavin resulted in a statistically significant ~10-fold higher lipid peroxidation rate compared with unirradiated control cultures and ~4–5-fold higher lipid peroxidation levels than detected in cell cultures irradiated in the absence of riboflavin (Figure 5A).

Concerning the impact of blue light exposure of the three urothelial tumour cells on fundamental mitochondrial respiratory chain parameter, the results displayed in Figure 5 show a strongly significant decrease in basal activity (Figure 5B), ATP production (Figure 5C), maximum respiration (Figure 5D) and spare respiration capacity (Figure 5E). Even if the control values of the individual tumour cell lines varied greatly among themselves, the modulation of the parameters shown by the light exposure was nevertheless comparatively equally pronounced in the three tumour cell lines.

## 3. Discussion

A major problem in the treatment of patients with high-risk non-muscle invasive bladder cancer (NMIBC) and high risk of tumour progression from T1 to T2 and greater is BCG refractoriness. For patients who have a BCG-refractory tumour, the guidelines require a radical cystectomy. If NMIBC develops from a BCG-refractory high-risk NMIBC, this tumour is refractory to chemotherapy and has lymph node involvement and/or organ-transcending growth at the time of surgery. Previous studies in which patients were offered an alternative to cystectomy have so far been less successful [43,44]. It is therefore important to find new options for bladder-preserving instillation therapy for these patients. Combination therapies could be an alternative to traditional chemotherapy-only treatments, potentially reducing the dose of the toxic substance, thereby reducing unwanted side effects and burdens. With increasing advances in the field of laser and LED technology, light in its entire spectrum, from ultraviolet to deep infrared, represents an interesting, specific and effective medium for combination therapies [45,46,47]. The therapy-improving potential of light applications, possibly also in combination with suitable photosensitisers, is based on the light-induced generation of reactive oxygen species (ROS), which contributes synergistically or additively to increasing the cytotoxic potential of a chemotherapeutic agent by increasing the intracellular or extracellular oxidative stress [48,49,50,51]. Thus, increasing the therapeutic effect by using photodynamic therapy methods represents one such new therapeutic option. Preliminary data from initial clinical studies demonstrate that photodynamic therapy using the photosensitiser TLD-1433 followed by activation with a 520 nm intravesical laser is an effective and viable treatment option for patients who do not respond to BCG [52].

Recently, we were able to show that blue light with a wavelength of 453 nm, used in non-toxic doses, significantly increases the cytotoxic potential of mitomycin C against the human bladder cell carcinoma cell line RT-112. In that study, light-modulated increase in cytotoxicity, mainly registered in the form of apoptosis and secondary necrosis, correlated with a significant increase in intracellular production of ROS and a decrease in mitochondrial respiration [35].

The primary therapeutic agent used here was gemcitabine, which shows advantages in side effect profile and a better efficacy compared with standard installations therapy with MMC [53,54]. Furthermore, clinical data show that gemcitabine has a good tumour response [54], so that in BCG-pre-treated patients, intravesical gemcitabine therapy after trans urethral resection of bladder tumour (TURBT) could be a good alternative therapy of non-muscle-invasive bladder cancer. Although, in some studies, the cell lines used in the present study, BFTC-905 [55] and RT-112 [56], were described as gemcitabine resistant, whereas SW-1710 [57] cells showed a good sensitivity to gemcitabine, our results indicate that all three cell lines examined showed comparable good sensitivity to gemcitabine. In the concentration interval of 10 to 50 ng/mL, gemcitabine induced an average cytotoxicity rate of 10 to 60% after an incubation time of 48 h. This therapeutic potential was in line with our expectations, since the concentrations used by us largely corresponded to the peak plasma concentrations, 5 to 320 µM, found in patients treated with gemcitabine at therapeutic relevant doses of 800–5700 mg/m^2^ [58].

In the following, we focused on the experiment variant with 10 ng/mL gemcitabine. At this gemcitabine concentration we observed no or only a weak toxicity, which could be significantly increased by exposure to non- or only slightly toxic doses of blue light. With 10 ng/mL gemcitabine and blue light, the toxicity rate in the treated cell cultures was approximately 50% and thus corresponded to cytotoxicity rates that were achieved with 25 ng/mL gemcitabine alone. This means that even with this measure, an identical therapeutic result could be achieved with a 60% lower exposure to the chemotherapeutic agent. These increases in cytotoxicity were paralleled by a 2–4-fold increase in intracellular reactive oxygen species (ROS) formation, an essential factor and mediator of cell toxicity.

The biological outcome of increased ROS production elicited by exposure to blue light is fully consistent with the outcome of any other increased ROS exposure. This includes a reduction in the migration, proliferation and differentiation capacity of cells, and, exceeding a critical threshold value, high ROS production can severely impair cell vitality up to and including a pronounced cytotoxic effect [59,60,61,62]. Under physiological conditions, all cell types are able to generate ROS. Physiological ROS generation by enzymes like flavoenzymes of the mitochondrial respiratory chain, isoenzymes of the NADPH oxidase family (NOX), 5-lipoxygenase or xanthine oxidoreductase (XOR) [63] represents important cellular control systems in immune processes, the regulation of cell growth, cell differentiation and various cell signalling pathways [64,65]. These enzyme families generate ROS by enzyme-controlled transfer of electrons from reduced flavin residues to oxygen (O_2_), with superoxide radical anions (O_2_^−.^) and hydrogen peroxide (H_2_O_2_) being formed as primary and secondary products, respectively. The ROS synthesis is strongly dependent on the availability of NADH or NADPH, which act as electron donors for the reduction of the flavin residues [66,67]. Interestingly, flavin residues can also be reduced by blue light, completely without the participation of NADH or NADPH [59]. After photoreduction, due to a process of an intramolecular light-independent flavin reoxidation, analogous to physiological conditions, electrons are transferred to O_2_, which might result in ROS formation [62,66,68,69]. Thus, the extent of blue light-induced ROS production is a function of the light dose, but to a greater extent a function of the amount of photo-reduced flavin residues present. As we show here, exogenously applied flavin derivatives at a given light dose increase the ROS synthesis induced by blue light disproportionately and also the corresponding biological effects. Thus, in the presence of riboflavin (10 µM), subsequent irradiation of cell cultures with blue light also leads to a significant increase in cell cytotoxicity in the exposed cell cultures of approx. 50%, a value that we also achieve with 10 ng/mL gemcitabine and blue light. This result is consistent with previous results showing that riboflavin effectively generates ROS under the influence of blue light and thus might significantly contribute to cytotoxicity [70,71,72,73].

The use of a combination of the three effect parameters examined here, gemcitabine, blue light and riboflavin, leads to further additive cytotoxic effects of the three tumour cell lines. With 10 ng/mL gemcitabine, 10 µM riboflavin and a moderate dose of blue light, we achieve a cytotoxic potential in the urothelial cancer cell line cultures that could only be observed with the sole use of gemcitabine at a concentration of 50 ng/mL. It seems that the immensely increased cytotoxicity is the result of different molecular mechanisms. On the one hand, there is the well-known interaction between gemcitabine and the DNA, which induces a basic cytotoxic sensitivity of the cells. This is graded and significantly enhanced by increased ROS generation induced by blue light and riboflavin. The result of this high ROS load on cells might be the loss of integrity and function of nucleic acids, proteins and membranes. As we were able to show earlier [35], the latter point in particular can lead to the escape of cytochrome C into the cytosol and thus to the induction of apoptotic cell death by disrupting the membrane function of mitochondria. The molecular mechanism of apoptosis is fine-tuned and programmed by the cell. The need for energy in the form of ATP is characteristic of apoptosis. Any form of ATP deficiency leads to the termination of the programmed cascade of events of apoptosis at the appropriate point in the mechanism and leads to the necrosis of the cell, which, in such a case, is referred to as secondary necrosis [74]. Indeed, secondary necrosis accounts for a significant proportion of blue light-induced cell death [35]. An ATP deficiency situation, which could promote the formation of secondary necrosis, can be achieved when a cell cannot produce ATP, for example due to impaired glycolysis or mitochondrial respiration or due to the unavailability of the substrates required for ATP production. The results of the current study clearly indicate that, despite all the differences between the three tumour cell lines examined, blue light significantly decreases mitochondrial basal activity and ATP production. The reduction in cellular ATP production is thus another mechanism by which blue light contributes to increasing the tumour cytotoxic potential of gemcitabine.

Of course, the question arises regarding the selectivity of the proposed therapy option and potential side effects and how to minimize them. Concerning the proposed therapy option, we can certainly exclude additional systemic side effects beyond those attributable solely to gemcitabine treatment. However, there is indeed a real possibility that, due to the described combination therapy, damage could occur locally, directly in the area of treatment, to healthy tissue of the treated or immediately adjacent tissue. The relevant parameter that could induce local cytotoxicity and potential side effects compared with the sole action of gemcitabine is the blue light, whose cytotoxic effect is significantly enhanced in the presence of riboflavin. An increase in toxicity due to the sole use of riboflavin without the light impact can, however, be ruled out. As mentioned above, the toxic effect of blue light is mediated by flavins by modulating and increasing the generation of reactive oxygen species (ROS). How could specificity of the therapy option described here be achieved or optimised, or how could side effects be minimised?

As shown in Figure 2, we observed no significant increase in cytotoxicity in the investigated tumour cell lines with blue light irradiation up to a dose of 110 J/cm^2^. However, when using light doses above this threshold, cytotoxicity increased dose-dependently. There are reports indicating that blue light of identical wavelength showed no significant toxicity on human fibroblasts up to 300 J/cm^2^ and even up to 500 J/cm^2^ on human keratinocytes. These findings suggest a potential selectivity regarding the sensitivity of different cell types. The cytotoxic effect of blue light depends on the level of intracellularly generated ROS such as O_2_^−.^ and H_2_O_2_ and the antioxidant capacity of the cell mediated by the corresponding specific ROS-scavenging enzymes superoxide dismutase (SOD) and catalase, respectively. The knowledge of these parameters could potentially be utilised for selectively enhancing the toxicity in tumour tissue while simultaneously aiming for minimal phototoxicity in adjacent healthy cell types. Furthermore, the penetration depth of blue light into biological tissue is approximately 1–2 mm, making light-based therapy particularly suitable for the treatment of non-invasive superficial tumours. By locally applying the photosensitiser only to the area of the tumour tissue and using, for example, a template to define a sharply delineated irradiation area, undesired local side effects could be largely minimised or even prevented.

In summary, our results clearly indicate that the gemcitabine dosage can be reduced if combined with blue light and riboflavin. If transferable to clinical practice, this could be an improvement for bladder carcinoma therapy by reducing the gemcitabine dosage and thus, simultaneously, the side effects for the patients. Of course, there are limitations. The penetration depth of blue light in biological tissue is about 1.5 mm. Such treatment of deeply invaded tumours would only make limited sense. Clinical application supported by blue light would be limited to the treatment of superficial, epithelial and freely accessible tumours or tumours accessible with the help of special medical technology. In the case of bladder cell carcinoma, the existing modern endoscopy technology could be used. Such a light-assisted therapy option could be individually adapted to the respective patient through the use of higher light doses and further promote the success of the therapy. By using special templates that sharply delimit the area to be treated, healthy areas in the treatment area could also be protected from possible side effects of the therapy.

## 4. Materials and Methods

### 4.1. Materials

If not otherwise indicated, all chemicals, antibodies and assay kits were purchased from Sigma-Aldrich Chemie GmbH (Munich, Germany).

### 4.2. Cell Lines

The three adherently growing urothelial bladder cancer cell lines used here, BFTC-905, RT-112 and SW-1710, were purchased from Leibniz Institute DSMZ-German Collection of Microorganisms and Cell Cultures GmbH (Braunschweig, Germany).

BFTC-905 cells were established from a 51-year-old woman with a grade G3 papillary transitional cell carcinoma of the urinary bladder. In the cell culture, these cells grow in form of a monolayer with a doubling time of 60 to 70 h [75]. RT-112 cells were established from the transitional cell carcinoma (histological grade G2) excised from a woman with untreated primary urinary bladder carcinoma. In the cell culture, RT-112 cells grow in monolayers with a doubling time about 35 h [76]. SW-1710 cells were established from the bladder tumour of an 84-year-old woman following transurethral tumour resection; they were mesenchymal cells of a histological grade G3 bladder carcinoma. In the cell culture, they grow elongated and in monolayers with a doubling time of 25 to 32 h [77]. These three cell lines are representative of a broad spectrum of potential urothelial bladder cancer phenotypes, including epithelial and mesenchymal subtypes [78].

### 4.3. Medium and Cell Culture

The RPMI 1640 medium (Life Technologies Ltd.; Paisley, UK) we used to cultivate cell lines contained 1 g/L glucose (Life Technologies Ltd.; Paisley, UK), 1% penicillin/streptomycin (PAN-Biotech GmbH; Aidenbach, Germany), 5% heat inactivated foetal bovine serum (Fetal Bovine Serum Gold, PAA Laboratories GmbH; Cölbe, Germany) and 1% sodium pyruvate (Life Technologies Ltd.; Paisley, UK). The cells were cultivated as adherent monolayers in T175 culture flasks (CELLSTAR^®^ Cell Culture Flasks 175 cm^2^ red filter cap, Greiner Bio-One GmbH; Kremsmünster, Austria) and were maintained at 37 °C in a humidified atmosphere with 5% CO_2_ (Incubator; Thermo Fisher Scientific GmbH; Dreieich, Germany).

The cells were inspected daily with a light microscope, and after reaching a confluence of 70–80%, they were sub-cultured. The cells were detached by incubating the cultures with 1% trypsin/EDTA solution (0.05%/0.02%) in PBS without Ca^2+^ and Mg^2+^ (PAN-Biotech; Aidenbach, Germany) for 5 min at 37 °C. Trypsin activity was inhibited by a trypsin inhibitor (Life Technologies Ltd.; Paisley, UK). After trypsin neutralisation, the cells were relieved by a cell scraper, transferred into a 25 mL tube (FALCON, Corning Science Mexico S.A. de C.N Avenida Industrial del nord SIN) and centrifuged (Heraeus Megafuge 16 R, Thermo Fisher Scientific GmbH; Dreieich, Germany) with spinning at 300× *g* for 5 min. Then, the supernatant was removed and the cells were resuspended in culture medium and incubated at 37 °C in T175 culture flasks. Under the usage of the Neubauer chamber, the cell numbers were determined. The medium was changed every two or three days, and, once a month, parts of the cells were cryo-conserved. The cell culture was cultivated in a humidified incubator (95% air and 5% CO_2_ at 37 °C).

### 4.4. Light Source

For our experimental setup, we used a narrow-band LED-device (10 cm × 12 cm LED-array with 60 LEDs), whose irradiance was characterised in advance with an integrating (Ulbricht) sphere. The electrical and optical device parameters were as follows: I = 1.75 A, U = 40 V_DC_, P_in_ = 69 W, optical output power = 13 W, LED maximum intensity was 0.21 W/nm at 453 nm and emission spectrum was 453 ± 10 nm. LED arrays used for this purpose were supplied by Philips Research (Aachen, Germany). During the irradiation process, the distance between the LED surface and the cell monolayer was 5 cm, and the radiation power was 39 mW/cm^2^. Irradiation time was 2820 s, resulting in a fluence of 110 J/cm^2^. Under the conditions of the light exposure described, the sample temperature never exceeded 33 °C. In order to create comparable conditions, the non-irradiated control plates were maintained for 45 min in a 33 °C warm windowed heating cabinet. Control experiments that investigated the dehydration effects of light exposure revealed that osmotic effects could be negligible.

### 4.5. Chemotherapy

As chemotherapy, we used gemcitabine, a cytidine analogue. Gemcitabine interferes with the DNA synthesis by inhibiting the ribonucleotide reductase. This cytotoxic drug was acquired from the pharmacy of the Heinrich-Heine-University in Düsseldorf, which is part of the same organisation as our laboratory. The concentration of gemcitabine in the acquired solution was 40 mg/mL, and the molar mass of gemcitabine was 263.198 g/Mol. In experiments, gemcitabine was used at concentrations of 10, 25 and 50 ng/mL.

### 4.6. Riboflavin

Riboflavin is a part of the vitamin B2 complex and, in the form of the flavin adenine dinucleotide (FAD) of the respiratory chain, it serves as an important component of the metabolism in living organisms. Here, we used riboflavin as the photosensitiser and source of reactive oxygen species (ROS) generation. The molar mass of riboflavin is 376.36 g/mol, and we used this compound at concentrations of 10 or 12.5 μM.

### 4.7. Experimental Setup

Cells were transferred to transparent 12-well culture plates (12-well CELLSTAR^®^ Tissue Culture Plates, Greiner Bio-One GmbH; Kremsmünster, Austria) with 3 × 10^4^ cells per well and cultivated in culture medium overnight at 37 °C and 5% CO_2_ to achieve adherence and a nearly confluent monolayer. The respective cell cultures were then incubated with gemcitabine in the specified concentrations for 24 h, washed twice with PBS, irradiated with blue light (453 nm, 110 J/cm^2^) in the presence or absence of riboflavin (10 or 12.5 μM), washed again with PBS and incubated with gemcitabine for additional 24 h at 37 °C and 5% CO_2_. Finally, cells, cell culture supernatants or cell lysates were collected for further analyses.

It is important to note that the culture medium was replaced with PBS prior to each exposure to blue light (2 mL/well for 6-well plates, 1 mL for 12-well plates, 0.5 mL for 24-well plates, 200 µL for 96-well plates) and that the irradiation of cells was carried out in PBS.

### 4.8. MTT Assay

For the MTT-cell viability assay, 1.5 × 10^3^ cells per well were plated into a 96-well plate, incubated with 100 μL/well MTT solution (0.5 mg/mL; thiazolyl blue tetrazolium bromide) at 37 °C in the dark for 2 h. Then, the supernatant was discarded, the cells were lysed by addition of 150 μL of DMSO for 10 min, and 100 µL of this solution was transferred to a new 96-well microtiter plate. Finally, an absorbance at 540 nm was detected using the VICTOR multilabel counter.

### 4.9. Cell Titer Blue Viability Assay

With this assay, we measured the cell viability and the activity of the respiratory chain. The metabolisation of resazurin to resorufin (CellTiterBlue, Promega, Madison, WI, USA) was captured and, with this, the metabolic activity of the cells. For the CellTiterBlue assay, we plated 3 × 10^4^ cells per well and performed the assay as prescribed in the protocol. Thus, the cells were incubated for sixty minutes with the CellTiterBlue reagent (1 h;1:20 dilution with medium; 400 µL). After this, 2 × 100 μL of the medium/CellTiterBlue reagent solution from each well was transferred to a microtiter plate from each well. The fluorescence spectrometer (Ex/Em = 540/590 nm; VICTOR II, Perkin Elmer, Waltham, MA, USA) was used afterwards to measure cell viability.

### 4.10. Immunofluorescence

With the help of fluorescence microscopy and three different dyes, we were able to differentiate between living cells and cells undergoing apoptosis or necrosis. For that, the cell cultures were washed with PBS (PBS without Ca^2+^, Mg^2+^) and then incubated for five minutes with a dye composition of Hoechst 33342 dye (H33342, 0.5 µg/mL), fluorescein diacetate (FDA, 2 µg/mL) and propidium iodide (PJ, 0.5 µg/mL). Afterwards, the cells were again washed with PBS. FDA can be detected as a green fluorescent colouring of living cells [79], H33342 stains the chromatin of living cells by blue fluorescence and thus allows to evaluate nuclear morphology of apoptotic cells, whereas PJ only penetrates “leaky” membranes and thus is an indicator for necrotic cells, which glow in red fluorescence [79]. The fluorescence signals were visualised using a Zeiss fluorescence microscope (H33342: excitation: 355 nm, emission: 465 nm; propidium iodide: excitation: 520 nm, emission: 610 nm). For further evaluation and analysis, image software ImageJ Version 1.54a (National Institutes of Health (NIH), Bethesda, MD, USA) was used.

### 4.11. Lipid Peroxidation

Lipid peroxidation (LPO) was determined in non-treated tumour cell cultures (2 × 10^7^ cells) or cultures irradiated with blue light (110 J/cm^2^) in the absence or presence of the respective additives using the lipid hydroperoxide (LPO) assay kit (Cayman Chemical Company, Ann Arbor, MI, USA). The LPO assay kit measures the hydroperoxides directly utilising the redox reactions with ferrous ions [80].

### 4.12. Cell Preparation for Seahorse Assay

Mitochondrial respiration oxygen consumption rates (OCR), which allow conclusions about different parameters of mitochondrial respiration, were recorded by using an extracellular flow analyser Agilent Seahorse XF24 (Seahorse Bioscience, North Billerica, MA, USA) [81] and the Mito Stress Test Assay (Seahorse Bioscience, North Billerica, MA, USA), which was performed according to the protocol described by Butler et al. [82]. Additionally, using the same instrument together with the company’s Extracellular Acidification Rate (ECAR) Assay, we studied the extracellular acidification rates of the respective cell cultures as indicators of energy-producing pathways of glycolysis [83]. The analyses of the OCR and ECAR of the untreated and light-exposed tumour cell cultures used were carried out under identical conditions, as previously described by us [35].

### 4.13. Statistical Analysis

All values were reported as means ± SD and derived from the indicated number of independent experiments. Statistical analyses were conducted with GraphPad Prism 8.0. Data were analysed using the two-way ANOVA test. Differences between the independent variables were checked in post hoc tests and Tukey’s honestly significant difference (HSD) tests for variables. Data were also analysed using the paired *t*-test. A *p*-value < 0.05 (*p* < 0.05) was considered to be significant.

## 5. Summary of Results

Gemcitabine exhibited concentration-dependent cytotoxicity.Blue light exposure resulted in dose-dependent cytotoxicity in all cell lines.Riboflavin enhanced blue light-induced cytotoxicity.Combined treatment of gemcitabine, blue light and riboflavin showed synergistic cytotoxicity.Blue light exposure increased lipid peroxidation and decreased mitochondrial respiratory chain activity.

These results indicate a potential synergistic cytotoxic effect of combined blue light-activated riboflavin and gemcitabine on urothelial carcinoma cells, possibly through modulation of lipid peroxidation and mitochondrial function.

## Figures and Tables

**Figure 1 ijms-25-04868-f001:**
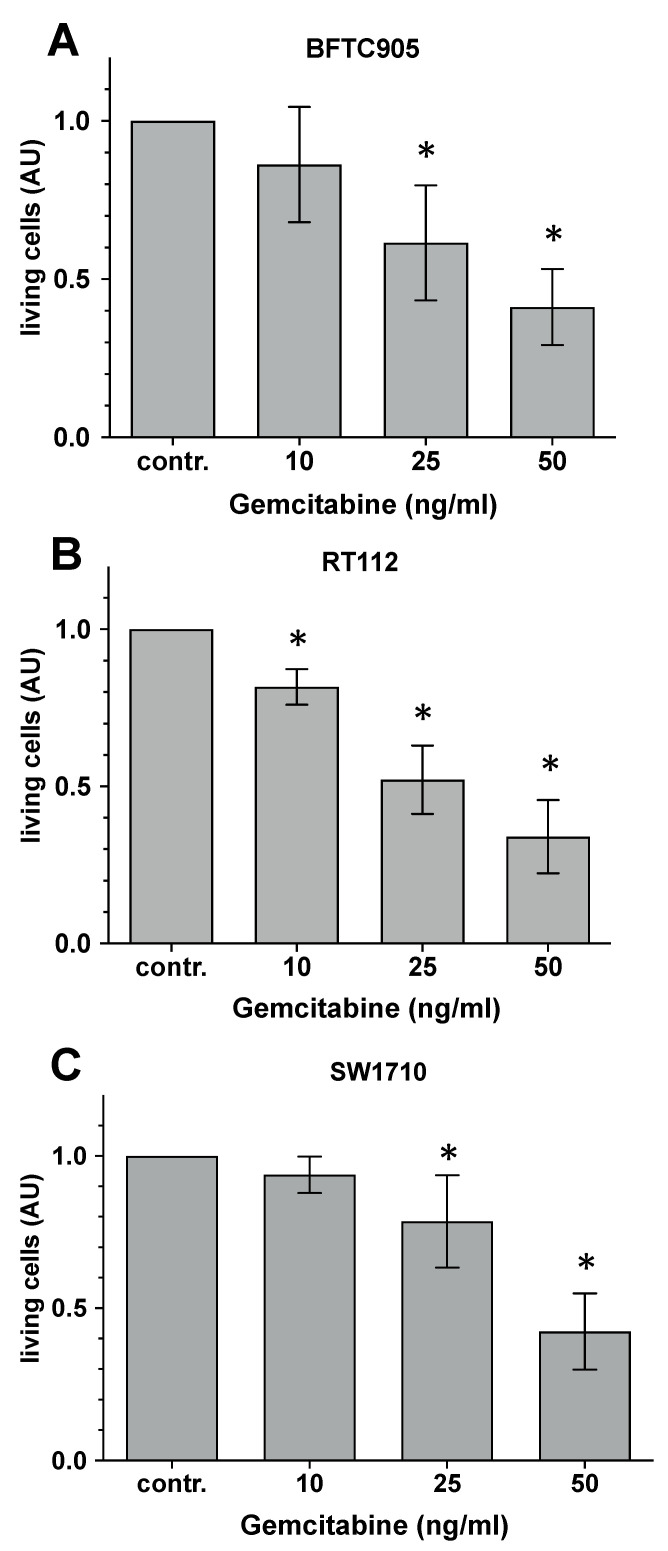
Cytotoxic capacity of gemcitabine. Urothelial carcinoma cell line cultures, (**A**) BFTC-905; (**B**) RT-112; (**C**) SW-1710 were incubated with 10, 25 or 50 µM gemcitabine for 48 h. The relative number of living cells was quantified by CTB assay. Bars represent the mean ± S.D, normalised to the value of the non-treated controls (contr.) of eight individual experiments (n = 8). Statistical evaluation (two-way ANOVA) was carried out with the values of the original measurements, i.e., before the normalisation process to the respective control value shown here (contr.). *, *p* < 0.05 compared with the respective control cultures maintained in the absence of gemcitabine.

**Figure 2 ijms-25-04868-f002:**
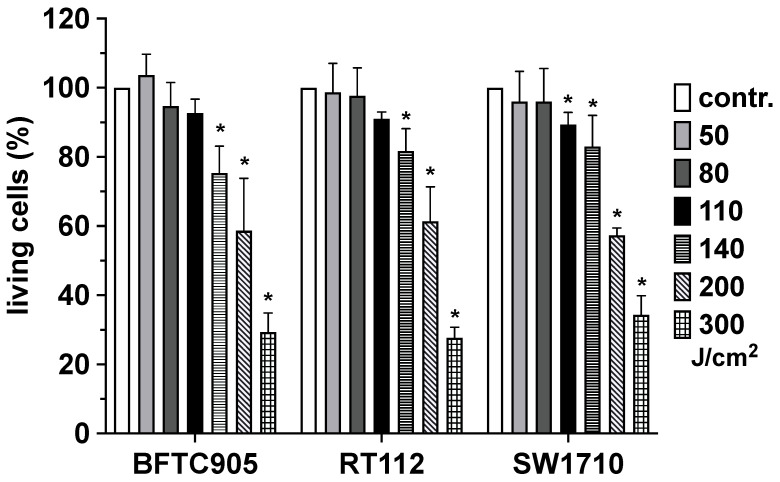
Impact of blue light exposure on viability of urothelial carcinoma cells. Urothelial carcinoma cell line cultures (BFTC-905; RT-112; SW-1710) were irradiated with blue light (l = 453 nm) at light doses indicated (50, 80,110, 140, 200, 300 J/cm^2^). Twenty-four hours after light exposure, the number of living cells was detected by MTT assay. Bars represent the mean ± S.D, normalised to the value of the non-treated controls of four individual experiments (n = 4). Statistical evaluation (paired *t*-test) was carried out with the values of the original measurements, i.e., before the normalisation process to the respective control value shown here (contr.). *, *p* < 0.05 compared with the respective non-irradiated cell cultures.

**Figure 3 ijms-25-04868-f003:**
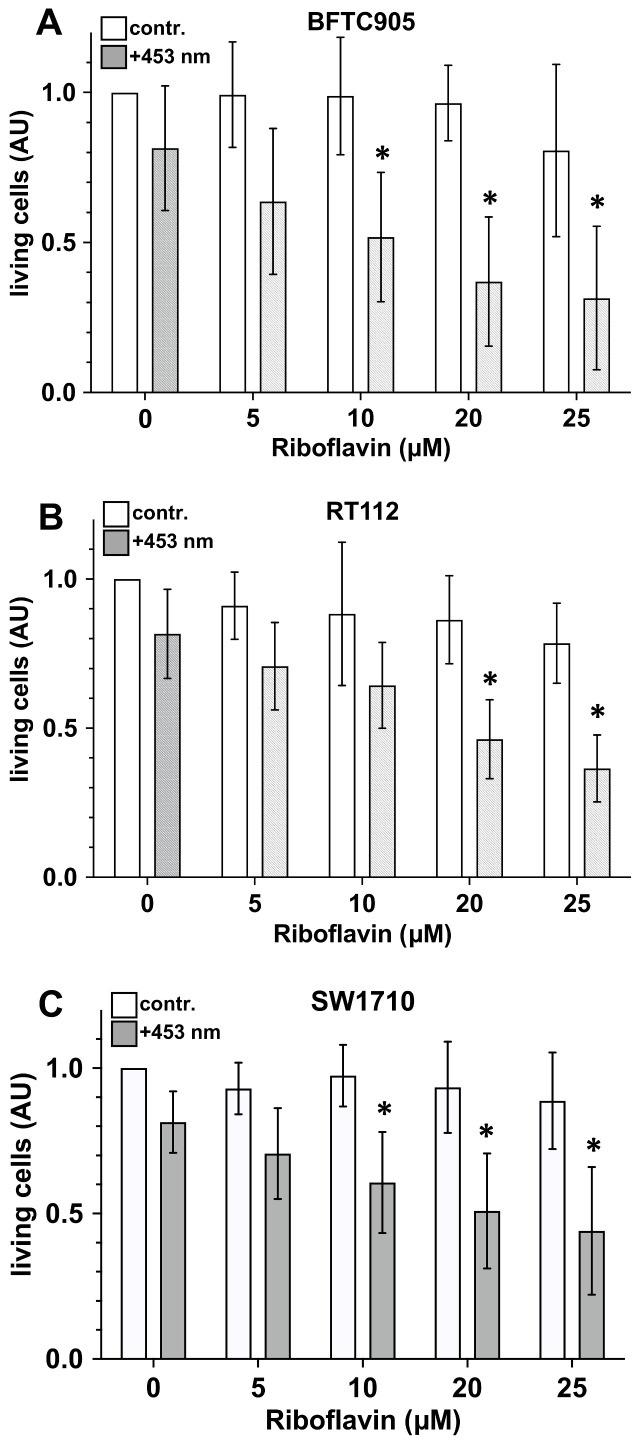
Impact of blue light-activated riboflavin on viability of exposed urothelial carcinoma cell cultures. Urothelial carcinoma cell line cultures (**A**) BFTC-905; (**B**) RT-112; (**C**) SW-1710 were exposed to blue light (grey bars, 110 J/cm^2^) in the absence (0 µM) or presence of 5, 10, 20 or 25 µM riboflavin. White bars represent the values of the respective non-irradiated control cultures. Twenty-four hours after light exposure, the number of living cells was detected by MTT assay. Bars represent the mean ± S.D, normalised to the value of the non-treated controls of eight individual experiments (n = 8). The statistical evaluation (two-way ANOVA) was carried out with the values of the original measurements, i.e., before the normalisation process to the respective control value shown here (contr.). *, *p* < 0.05 compared with the value of the respective non-irradiated control cultures.

**Figure 4 ijms-25-04868-f004:**
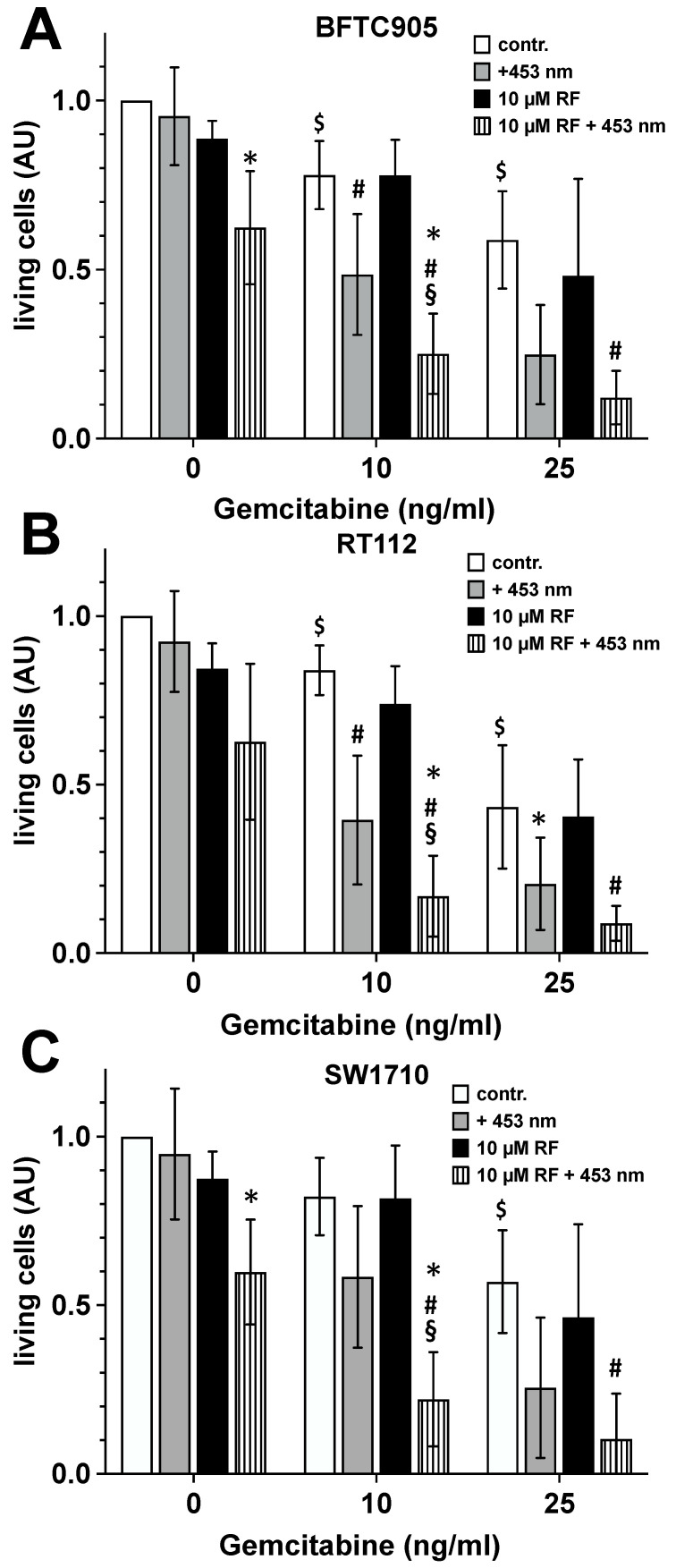
Impact of blue light-activated riboflavin on the cytotoxic capacity of gemcitabine. Urothelial carcinoma cell line cultures ((**A**) BFTC-905; (**B**) RT-112; (**C**) SW-1710) were maintained in the absence or presence of 10 or 25 ng/mL gemcitabine (contr., white bars). Additionally, these cultures were incubated with 10 µM riboflavin (10 µM RF, black bars). Finally, control cultures (contr., white bars) were exposed to blue light with a dose of 110 J/cm^2^ (+453 nm, grey bars,) and the riboflavin-containing cultures (10 µM RF, black bars) also were exposed to blue light with a dose of 110 J/cm^2^ (10 µM RF + 453 nm, lined bars). Twenty-four hours after light exposure, the number of living cells was detected by MTT assay. Bars represent the mean ± S.D, normalised to the value of the non-treated controls, of seven individual experiments (n = 7). The statistical evaluation (two-way ANOVA) was carried out with the values of the original measurements, i.e., before the normalisation process to the respective control value shown here (contr.). *, *p* < 0.05 compared with the respective cultures irradiated in the absence of riboflavin (grey bars); #, *p* < 0.05 compared with the respective control cultures (contr., white bars); §, *p* < 0.05 compared with the respective non-irradiated riboflavin-containing cultures (10 µM RF, black bars); $, *p* < 0.05 compared with the respective control cultures maintained in the absence of gemcitabine (white bars, 0 ng/mL gemcitabine).

**Figure 5 ijms-25-04868-f005:**
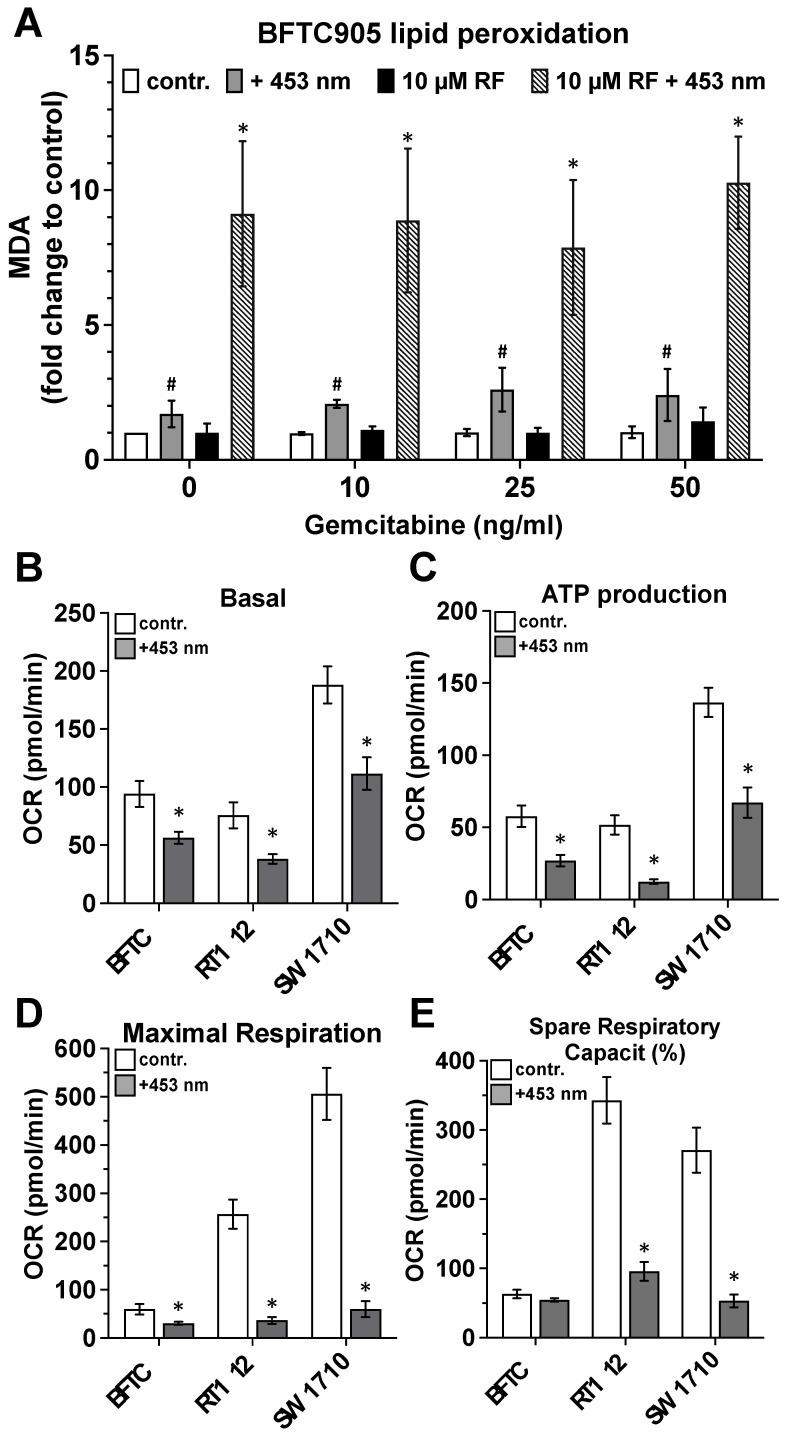
Impact of blue light-activated riboflavin on lipid peroxidation and mitochondrial respiratory chain activity of urothelial carcinoma cell cultures. (**A**), Urothelial carcinoma cells (BFTC-905) were maintained in the absence or presence of 10, 25 or 50 ng/mL gemcitabine (contr., white bars). Additionally, these cultures were incubated with 10 µM riboflavin (10 µM RF, black bars). Finally, control cultures (contr., white bars) were exposed to blue light with a dose of 110 J/cm^2^ (+453 nm, grey bars,) and the riboflavin-containing cultures (10 µM RF, black bars) also were exposed to blue light with a dose of 110 J/cm^2^ (10 µM RF + 453 nm, lined bars). Twenty-four hours after light exposure, the concentration of malondialdehyde as a lipid peroxidation marker was quantified using a specific assay. Bars represent the mean ± S.D, normalised to the value of the non-treated controls, of three individual experiments (n = 3). The statistical evaluation (paired *t*-test) was carried out with the values of the original measurements, i.e., before the normalisation process to the respective control value shown here (contr.). *, *p* < 0.05 compared with all other cultures; #, *p* < 0.05 compared with the respective control cultures (contr., white bars). Additionally, key parameters of the respiratory chain, basal respiration (**B**), ATP production rate (**C**), maximal respiration (**D**) and relative spare respiration capacity (**E**) of non-irradiated (white bars) as well as blue light exposed (BL, 110 J/cm^2^) BFTC-905, RT-112 or SW-1710 tumour cell cultures were detected. The statistical evaluation was carried out with the values of the original measurements. *, *p* < 0.05 compared with the respective values obtained with the non-irradiated cultures.

## Data Availability

The datasets used and analysed during the current study are available from the corresponding author on reasonable request.

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
