# Peer review of "Exposure of Bladder Cancer Cells to Blue Light (λ = 453 nm) in the Presence of Riboflavin Synergistically Enhances the Cytotoxic Efficiency of Gemcitabine"

_ijms, 2024, doi:10.3390/ijms25094868_

Round 1

Reviewer 1 Report

Comments and Suggestions for Authors

The manuscript explores a novel approach to enhancing the therapeutic efficacy of gemcitabine in non-muscle invasive bladder cancer through the use of blue light and riboflavin. The findings are intriguing and have the potential to reduce the therapeutic burden associated with gemcitabine treatment. However, there are several aspects that need clarification and further investigation.

Specific Comments:

1. The choice of a 24-hour incubation period with riboflavin requires justification. Have the authors considered or tested shorter incubation times? It would be beneficial to know if a more convenient and feasible treatment process could be achieved without compromising the therapeutic efficacy.

2. The authors mention that BFTC-905 and RT-112 cell lines are resistant to gemcitabine, while SW-1710 cells are sensitive. However, Figure 1 does not show a significant difference in gemcitabine-induced cytotoxicity between SW-1710 and the other two cell lines. This needs to be addressed and explained, as it may affect the interpretation of the results.

3. The authors need to consider and discuss the potential side effects of their proposed treatment on the bladder mucosa and surrounding organs. The risk of bleeding and damage to healthy tissues should be addressed. Additionally, the authors should provide more evidence or discussion on the specificity of the treatment towards tumor cells to ensure minimal off-target effects. There should be clearer evidence or discussion regarding the targeting specificity of the drugs.

Author Response

Reviewer #1

  1. The choice of a 24-hour incubation period with riboflavin requires justification. Have the authors considered or tested shorter incubation times? It would be beneficial to know if a more convenient and feasible treatment process could be achieved without compromising the therapeutic efficacy.

Replay:

This seems to be a misunderstanding. As described in Section "2.7. Experimental Setup," the riboflavin was only present in the cell cultures during light exposure. However, from previous experiments, we also know that longer incubations with riboflavin before or after light exposure showed no different effects than the effect of the experimental setup used here.

  1. The authors mention that BFTC-905 and RT-112 cell lines are resistant to gemcitabine, while SW-1710 cells are sensitive. However, Figure 1 does not show a significant difference in gemcitabine-induced cytotoxicity between SW-1710 and the other two cell lines. This needs to be addressed and explained, as it may affect the interpretation of the results.

Reply:

Starting with line 401, we mention in the Discussion: “…, clinical data show, that gemcitabine has a good tumor response [63], so that in BCG-pretreated patients intravesical gemcitabine therapy after trans urethral resection of bladder tumor (TURBT) could be a good alternative therapy of non-muscle-invasive bladder cancer.  

However, we also cite findings from individual studies that demonstrate varying responses of the cells we used to gemcitabine. However, we note at the same point that we did not observe or confirm these aforementioned differences in sensitivity of the cell lines used in our setup. “Although in some studies the cell lines used in the present study, BFTC-905 [64] and RT-112 [65] were described as gemcitabine resistant, whereas SW-1710 [66] cells showed a good sensitivity to gemcitabine, our results indicate that all three cell lines examined showed comparable good sensitivity to gemcitabine. In the concentration interval of 10 to 50 µM, gemcitabine induced an average cytotoxicity rate of 10 to 60 % after an incubation time of 48 hours. This therapeutic potential was in line with our expectations, since the concentrations used by us largely corresponded to the peak plasma concentrations, 5 to 320 µM, found in patients treated with gemcitabine at therapeutic relevant doses of 800 – 5700 mg/m2 [67].

  1. The authors need to consider and discuss the potential side effects of their proposed treatment on the bladder mucosa and surrounding organs. The risk of bleeding and damage to healthy tissues should be addressed. Additionally, the authors should provide more evidence or discussion on the specificity of the treatment towards tumor cells to ensure minimal off-target effects. There should be clearer evidence or discussion regarding the targeting specificity of the drugs.

Reply:

In the potential therapy option mentioned here, which involves the use of the chemotherapeutic agent, the photo acceptor riboflavin, and blue light, we can reasonably assume that there would be no additional systemic side effects expected beyond those anticipated solely as a result of gemcitabine treatment alone. However, there is indeed a real possibility that due to the described combination therapy, damage could occur locally, directly in the area of treatment, to healthy tissue of the treated or immediately adjacent tissue.

The relevant parameter that could induce local cytotoxicity and potential side effects compared to the sole action of gemcitabine is the blue light, whose cytotoxic effect is significantly enhanced in the presence of riboflavin. An increase in toxicity due to the sole use of riboflavin without the light-impact, however, can be ruled out. As we mention in our manuscript, the physiological effect of blue light is mediated by its interaction with flavin residues of proteins, thereby affecting the functionality of these proteins. The toxic effect of blue light is also mediated by flavins, but in this case, by modulating and increasing the generation of reactive oxygen species (ROS). How could specificity of the therapy option described here be achieved or optimized, or how could side effects be minimized?

As shown in Figure 2, we observe no significant increase in cytotoxicity in the investigated tumor cell lines with blue light irradiation up to a dose of 110 J/cm2. However, when using light doses above this threshold, cytotoxicity increases dose-dependently. There are reports indicating that blue light of identical wavelength showed no significant toxicity on human fibroblasts up to 300 J/cm2 and even up to 500 J/cm2 on human keratinocytes. The cytotoxic effect of blue light depends on the level of intracellularly generated reactive oxygen species (ROS) such as superoxide radicals (O2-.) and hydrogen peroxide (H2O2) and the antioxidant capacity of a cell mediated by the corresponding specific ROS-scavenging enzymes superoxide dismutase (SOD) and catalase, respectively. The knowledge of these parameters could potentially be utilized for selectively enhancing the toxicity in tumor tissue while simultaneously aiming for minimal phototoxicity in adjacent healthy cell types.

The penetration depth of blue light into biological tissue is approximately 1-2 mm, making light-based therapy particularly suitable for the treatment of non-invasive, superficial tumors. By locally applying the photosensitizer only to the area of the tumor tissue and using, for example, a template to define a sharply delineated irradiation area, undesired local side effects could be largely minimized or even prevented.

We now discuss the issue of possible adverse effects in the manuscript's discussion.

Reviewer 2 Report

Comments and Suggestions for Authors

The paper submitted for review entitled "Exposure of bladder cancer cells to blue light (l = 453 nm) in the presence of riboflavin synergistically enhances the cytotoxic efficiency of Gemcitabine" is interesting. After reading it, I ask that the authors answer the following questions or pore over certain parts of it.

1. BFTC-905 cells grow about twice as slowly as others. Don't the authors think it would be worth taking this into account when planning the study? At the same concentrations of compounds and amount of light, there are fewer BFTC-905 cells than the others, which means that the active concentrations are higher running cultures at the same time for all cells (e.g., 24h or 48h).

2) With each graph, the authors should state what statistical test was used.

3. What was the number of replicates of each measurement?

4. If an ANOVA type test was used, state what post-hoc test was used.

5. Figure 3: the authors do not provide information on what results were shown using white bars.

6. Figure 5A: was this study conducted only using BFTC-905 cells?

7. The authors are not consistent in naming, e.g. sometimes the name BFTC905 is used and sometimes BFTC-905 is used.

8. It is worth adding a section at the end of the paper summarizing the results obtained.

Author Response

Reviewer #2

  1. BFTC-905 cells grow about twice as slowly as others. Don't the authors think it would be worth taking this into account when planning the study? At the same concentrations of compounds and amount of light, there are fewer BFTC-905 cells than the others, which means that the active concentrations are higher running cultures at the same time for all cells (e.g., 24h or 48h).

Reply:

Indeed, this is an aspect we did not consider. One circumstance that could help mitigate the reviewer's argument is the experimental setup we used. All cell types were seeded at a density on the cell culture plate that led to the formation of a confluent monolayer upon cell adhesion to the plastic surface. We ensured the same initial conditions regarding the cell count used in the assay for all three cell lines. In our view, our results also do not provide any discernible evidence that the aspect raised by the reviewer significantly altered our data.

2) With each graph, the authors should state what statistical test was used.

Reply:

These details can now be found in the legends to the figures; we have now emphasized them more prominently in their respective locations.

  1. What was the number of replicates of each measurement?

Reply:

These details were already present in the legends to the figures; we have now emphasized them more prominently in their respective locations.

  1. If an ANOVA type test was used, state what post-hoc test was used.

Reply:

We have now added these details at the corresponding location in the text: two-way ANOVA test. Differences between the independent variables were checked in post hoc tests, Tukey's Honestly Significant Difference (HSD) tests for variables.

  1. Figure 3: the authors do not provide information on what results were shown using white bars.

Reply:

We have now supplemented this information at the corresponding location in the text. White bars represent the values of the respective non-irradiated cultures.

  1. Figure 5A: was this study conducted only using BFTC-905 cells?

Reply:

Yes, we conducted this test exemplarily only with the BFTC905 cell line.

  1. The authors are not consistent in naming, e.g. sometimes the name BFTC905 is used and sometimes BFTC-905 is used.

Reply:

For space reasons, we have generally presented the names of the cell types within the graphs without hyphens and with hyphens in the text. We would like to maintain this format if it appears acceptable to the reviewer. 

  1. It is worth adding a section at the end of the paper summarizing the results obtained.

Reply:

We have now added a summary of the results at the end of the manuscript.

Reviewer 3 Report

Comments and Suggestions for Authors

The manuscript in case, "Exposure of bladder cancer cells to blue light (l = 453 nm) in the presence of riboflavin synergistically enhances the cytotoxic efficiency of Gemcitabine" represents a preclinical study in the much needed area of bladder tumors. The authors investigated a photodynamic approach in which blue-light exposure activates both gemcitabine and riboflavin as a boost to a higher cytotoxic effect on bladder cancer cell lines. The authors demonstrated that this triple therapy approach provides best treatment results and also they provide the basis for a possible dose reduction of gemcitabine. However, throughout the text some issues need to clarified:

1. lines 14-16: "In order to reduce possible side effects and the burden of the drug, constant efforts are made to optimize the therapeutic potential of the therapeutic agents and thus reduce the therapeutic burden". ...what do you mean by therapeutic burden? I understand the idea but please rephrase for better clarity.

2. The same goes for "In the current study, we went one step further and examined whether the therapeutically supportive effect of blue light can be further increased by also enhancing the effect of blue light, for example through wavelength-specific photoacceptors, in our case of riboflavin" 

and 

"In the present study, we investigated in our preclinical study a photodynamic approach for the treatment of bladder cancer by the cytosine analogue gemcitabine with blue light at a wavelength of 453 nm and the flavin compound riboflavin".

and

"In our preclinival study in-vitro study with, we characterized in three molecular well-defined urothelial carcinoma cell lines (BFTC-905, SW-1710, RT-112) the influence..."

Frequent syntax errors, please rephrase!

3. line 85: "...may act as photosensitizer for blue light as well" Please provide reference for the photosensitizing ability of gemcitabine.

4. lines 85-87: "Further apart....exhibits a strong antineoplastic effect"...it's pretty obvious, that's the reason we're using gemcitabine in cancer therapy.

5. Figure 4:  when compared to control all 3 cell lines revealed cytotoxicity even in the presence of just RF (black bars) and gemcitabin, without blue light exposure.. Explanation from figure 4 confirms this difference as statistically significant. Please provide a possible reason for this increased cytotoxic effect that obviously occured just by adding RF.

6. line 385: ".....light-induced generation of reactive oxygen species". Literature data and especially the ones you cited (ref. 59 and 60) do not support the conclusion of "light-induced generation of ROS", as in all visible spectrum. Please stick to blue light.

7. line 480: "The treatment..." Recommend "Such treatment..."

Comments on the Quality of English Language

Frequent syntax errors in the first part of the manuscript. Please have it read by a native English colleague.

Author Response

Reviewer #3

  1. lines 14-16: "In order to reduce possible side effects and the burden of the drug, constant efforts are made to optimize the therapeutic potential of the therapeutic agents and thus reduce the therapeutic burden". ...what do you mean by therapeutic burden? I understand the idea but please rephrase for better clarity.

Reply:

We changed the text to: “To reduce potential side effects, continuous efforts are made to optimize the therapeutic potential of drugs, thereby reducing the effective dose and consequently the pharmacological burden of the medication”

  1. The same goes for "In the current study, we went one step further and examined whether the therapeutically supportive effect of blue light can be further increased by also enhancing the effect of blue light, for example through wavelength-specific photoacceptors, in our case of riboflavin" 

Changed in: ” In the present study, we investigated whether the therapeutically supportive effect of blue light can be further enhanced by the additional use of the wavelength-specific photosensitizer riboflavin

and 

"In the present study, we investigated in our preclinical study a photodynamic approach for the treatment of bladder cancer by the cytosine analogue gemcitabine with blue light at a wavelength of 453 nm and the flavin compound riboflavin".

We have deleted this redundant sentence

and

"In our preclinival study in-vitro study with, we characterized in three molecular well-defined urothelial carcinoma cell lines (BFTC-905, SW-1710, RT-112) the influence..."

Now changed in : “In our preclinical in vitro study, we characterized the influence of exposure to blue light in the presence of the photosensitizer riboflavin on the therapeutic efficacy of gemcitabine in three molecularly defined urothelial carcinoma cell lines (BFTC-905, SW-1710, RT-112).”

  1. line 85: "...may act as photosensitizer for blue light as well" Please provide reference for the photosensitizing ability of gemcitabine.

Reply:

We have deleted this statement, which is the result of a misunderstanding. It refers to a work that aimed to demonstrate how one could release the gemcitabine embedded in a photosensitive matrix using blue light..

  1. lines 85-87: "Further apart....exhibits a strong antineoplastic effect"...it's pretty obvious, that's the reason we're using gemcitabine in cancer therapy.

Reply:

We have modified this passage accordingly.

  1. Figure 4:  when compared to control all 3 cell lines revealed cytotoxicity even in the presence of just RF (black bars) and gemcitabine, without blue light exposure. Explanation from figure 4 confirms this difference as statistically significant. Please provide a possible reason for this increased cytotoxic effect that obviously occurred just by adding RF.

Reply:

In Figure 3, we demonstrate that riboflavin alone, in the absence of blue light at the concentrations used here, exhibits no toxicity. In Figure 1, we show that gemcitabine displays significant and expected toxicity towards the cell lines at concentrations above 25 ng/ml, with the RT112 cell line showing significant toxicity at concentrations as low as 10 ng/ml. In the combination of both substances (Figure 4), we observe that riboflavin + gemcitabine, in the absence of blue light, does not exhibit significantly increased toxicity compared to gemcitabine alone.

  1. line 385: ".....light-induced generation of reactive oxygen species". Literature data and especially the ones you cited (ref. 59 and 60) do not support the conclusion of "light-induced generation of ROS", as in all visible spectrum. Please stick to blue light.

Reply:

We have adjusted the quotes accordingly

  1. line 480: "The treatment..." Recommend "Such treatment..."

Reply:

Done

Round 2

Reviewer 3 Report

Comments and Suggestions for Authors

Good work

Comments on the Quality of English Language

Please have the text read by a native English speaker. Some syntax errors are still present

Author Response

Dear Reviewer,

We have submitted our manuscript to a native English speaker scientist for review. Following her suggestions, we have corrected the manuscript and would be delighted if it has now reached a quality that would allow for its publication.